# Inhibition of Factor XI: A New Era in the Treatment of Venous Thromboembolism in Cancer Patients?

**DOI:** 10.3390/ijms241914433

**Published:** 2023-09-22

**Authors:** Géraldine Poenou, Marco Heestermans, Ludovic Lafaie, Sandrine Accassat, Nathalie Moulin, Alexandre Rodière, Bastien Petit, Cécile Duvillard, Patrick Mismetti, Laurent Bertoletti

**Affiliations:** 1Therapeutic and Vascular Medecine Department, Saint-Etienne Universitary Hospital Center, F-42270 Saint-Priest en Jarez, Francececile.duvillard@chu-st-etienne.fr (C.D.);; 2INSERM, U 1059 SAINBIOSE, Jean Monnet University, Mines Saint-Étienne, F-42023 Saint Priest en Jarez, France; 3French Blood Establishement Auvergne-Rhône-Alpes, Research Department, F-42023 Saint-Etienne, France; 4Geriatry Department, Saint-Etienne Universitary Hospital Center, F-42000 Saint-Etienne, France; 5INSERM, CIC-1408, Saint-Etienne Universitary Hospital Center, F-42055 Saint Priest en Jarez, France; 6F-CRIN INNOVTE Network, F-42000 Saint-Etienne, France

**Keywords:** FXI inhibitors, cancer-associated thrombosis, venous thromboembolism, catheter-related thrombosis

## Abstract

Direct oral anticoagulants against activated factor X and thrombin were the last milestone in thrombosis treatment. Step by step, they replaced antivitamin K and heparins in most of their therapeutic indications. As effective as the previous anticoagulant, the decreased but persistent risk of bleeding while using direct oral anticoagulants has created space for new therapeutics aiming to provide the same efficacy with better safety. On this basis, drug targeting factor XI emerged as an option. In particular, cancer patients might be one of the populations that will most benefit from this technical advance. In this review, after a brief presentation of the different factor IX inhibitors, we explore the potential benefit of this new treatment for cancer patients.

## 1. Introduction

From the beginning of the 20th century to the present day, we have experienced significant advancements in antithrombotic drugs [1]. The first anticoagulants (vitamin K antagonists and heparins) non-specifically targeted the procoagulant serine proteases of coagulation [2,3]. Then, at the dawn of the 21st century, the development of oral medications specifically targeting thrombin (FIIa) and activated factor X (FXa), formerly referred to as DOACs for direct oral anticoagulants, improved the management of patients with indication for anticoagulant therapy [4,5]. Over time, these specific antithrombotics have become the first-line treatment for most thrombosis patients [6,7]. However, there remain situations, such as thrombosis associated with cancer, where these drugs can be ineffective with a risk of recurrence, can be overly effective with a risk of iatrogenic bleeding, or can simply be inappropriate (severe renal failure or impossible oral intake in palliative situations) [8,9]. Driven by the search for an antithrombotic that would guarantee efficacy, not inferior to current anticoagulants but with a greater safety, the inhibition factor XI (FXI) has emerged as a very interesting hypothesis [10]. FXI appears to play an auxiliary role in physiological hemostasis but a predominant role in pathological thrombosis [11]; see Figure 1.

**Figure 1 ijms-24-14433-f001:**
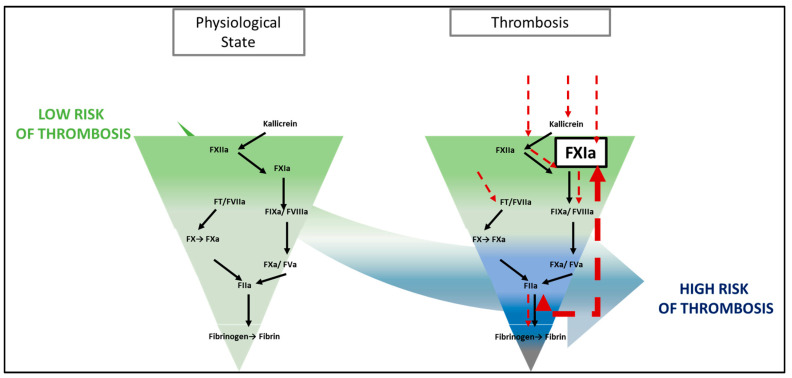
Factor XIa in the coagulation cascade. The coagulation cascade is represented as a pyramid where the closer you get to fibrin the higher is the risk of coagulation. In physiological state FXIa does not play a major role in the balance between bleeding and thrombosis. However, in cancer associated thrombosis several conditions activate FXI such as the cancer itself, surgery, catheter and inflammation. Black arrows represent the traditional pathway of coagulation. The red arrows represent the different pathway of activation of thrombosis in cancer associated thrombosis. Indeed, epidemiological data in patients with constitutional FXI deficiencies and experimental data in animals have strongly supported this hypothesis, with few prothrombotic events and almost no increased risk of bleeding in the central nervous system or gastrointestinal tract [12]. Phase II trials seem to confirm the interest in this strategy [13]. Parallel data on the pathophysiology of thrombotic events during cancer appear to support a prominent role of the contact path way via FXI in thrombus formation. Thus, the advent of FXI inhibitors, which have the potential to attenuate the response to prothrombotic stimuli with little or no disruption of hemostasis, could provide a safer anticoagulant [10]; see Figure 2.

Several FXI inhibitors are under development. These include the following: I/small molecules that bind to the active site of activated FXI (FXIa) and inhibit its activity (Asundexian and Milvexian), II/antibodies directed against FXI that inhibit the activation of FXI or the activity of FXIa (Osocimab, Abelacimab, and Xisomab), III/antisense oligonucleotides (ASOs, e.g., fesomersen) which reduce the synthesis of FXI by binding to the messenger RNA of FXI in hepatocytes and induce its degradation, and IV/natural inhibitors [13]. This review summarizes current progress and highlights how FXI inhibitors could be prescribed and used in the future treatment of patients with cancer-associated thrombosis.

## 2. General Information about FXI Inhibitors

### 2.1. The Inhibition of Contact Pathway as a New Therapeutic Target

In the classic portrayal of the coagulation cascade, which consists of the intrinsic, extrinsic, and common pathways, FXI is positioned within the intrinsic or contact activation arm of coagulation [14]. Given that deficiencies in contact activation proteins (like coagulation factor XII (FXII), prekallikrein (PK), and high-molecular-weight kininogen) are not associated with a bleeding tendency in humans, the role of contact activation in thrombin generation and plasma clotting was long considered separate from thrombosis, being seen primarily as a laboratory artifact, e.g., used to measure activated partial thromboplastin time (aPTT) tests. The aPTT test, which uses artificial contact activators like silica or ellagic acid to activate FXII and measure plasma clotting time, helps physicians identify anomalies concerning the plasma levels of certain coagulation factors [15]. Despite its deficiency causing a mild bleeding disorder known as hemophilia C, FXI was similarly deemed irrelevant for preventing venous thromboembolism (VTE), very similar to other contact factors [16]. However, recent times have seen a surge in interest in FXI, as it is now believed that it is more significant for thrombosis than for hemostasis. Accordingly, compounds inhibiting FXI function, like ASOs or blocking antibodies, reduced thrombosis in preclinical animal studies [16,17,18]. Recently, several contact activators linked to the onset of VTE, such as DNA, RNA, platelet polyphosphates, and misfolded proteins, have been suggested as physiological FXII activators [19,20,21]. According to data from these preclinical studies, FXI appears to serve solely as a substrate of FXIIa and is thus involved in coagulation only due to contact activation [22]. Nevertheless, this does not fully represent the human scenario where FXI deficiency, unlike FXII and PK, aligns with a bleeding phenotype and, unlike for FXI plasma levels, FXII and PK levels are not linked with the risk of VTE. The specific activation of FXI by thrombin in thrombotic events will be developed further [23]. Multiple therapeutic strategies for targeting the FXI protein have been suggested and the essential pharmacological attributes of each drug class are outlined in Table 1.

ASOs (fesomersen) and monoclonal antibodies (like Xisomab 3G3/AB023, Osocimab/BAY1213790, or Abelacimab/MAA868) have longer half-lives in humans of up to 30 days, facilitating less-frequent dosing, which could potentially improve patient adherence and compliance [24,25]. Small peptidomimetic molecules, such as Milvexian/BMS-986177 or BAY 2433334, are currently the sole drug class available for oral administration [26,27]. These compounds have a shorter half-life (one day versus weeks) and renal clearance. Natural inhibitors like Ir-CPI (*Ixodes ricinus* contact phase inhibitor) appear to be the least-viable therapeutic option due to their slow onset and offset, coupled with a relatively high risk of adverse side effects. The impact of antibodies and ASOs on FXI differ in their time to effect; monoclonal antibodies reduce FXI function within hours post administration, whereas the effect of ASOs only becomes noticeable after several weeks [13]. In addition, ASOs may cause thrombocytopenia [28]. This time difference in onset likely explains why the development of FXI-inhibiting monoclonal antibodies is currently more prevalent and why they would be more interesting for cancer patients. Finally, antibodies do not seem to suffer from drug–drug interactions, another crucial difference from direct oral anticoagulants.

### 2.2. FXI Inhibitor Monitoring

For patients with a congenital FXIa deficiency or an FXIa deficiency acquired from treatment, routine coagulation tests are not very relevant [29]. The prothrombin time is a reflection of coagulation initiated by the FVIIa/TF complex; thus, there is no expected interference from resulting low levels of FXIa in this assay. On the contrary, on those same patients, when using the aPTT, which explores factors belonging to the intrinsic pathway such as FVIII, FIX, FXII, and FXI, an elongation of the aPTT might be observed [29]. However, while the aPTT could be useful for detecting the presence of an FXI inhibitor, it cannot serve as a tool for assessing the effectiveness or safety in relation to the risk of bleeding from a congenital or acquired FXIa deficiency [30]. Indeed, FXI is not an initiator of coagulation in the aPTT test; thus, the effect of FXI inhibition on the test will vary for the same anti-FXI depending on the patient and between different anti-FXIs. As the FXI assay is a test based on the principle of the aPTT, the rate found will not be indicative. Finally, global coagulation tests such as thrombin generation, which evaluates both the initiation and propagation of coagulation, could be interesting to test FXI inhibitor levels [31]. However, their lack of standardization and their non-ubiquitous nature in all diagnostic laboratories are factors to be considered before envisioning routine use.

### 2.3. Antidote

Given that FXI inhibition can potentially disrupt the activation of TAFI (thrombin-activatable fibrinolysis inhibitor), tranexamic acid seems to be a powerful candidate in the prevention and treatment of bleeding in patients receiving FXI inhibitors [32]. It is commonly administered to FXI-deficient patients requiring surgical intervention or urgent procedures associated with a high risk of bleeding. In the event of severe bleeding with fesomersen, replacement with an FXI concentrate or fresh-frozen plasma may be beneficial [33]. This type of replacement therapy is likely to be insufficient with other FXI inhibitors. Instead, treatment with low doses of recombinant FVIIa, an approach successfully used in patients with congenital FXI deficiency, or with a prothrombotic complex concentrate, may be necessary [34]. Fortunately, recombinant FVIIa has not been required in the completed phase II trials so far, and patients have undergone major surgery such as kidney transplantation without excessive bleeding. Therefore, the need for reversal may be less with FXI inhibitors than with FXa or FIIa inhibitors. Among other antidotes on the market, andexanet alfa and idarucizumab, both highly specific, will assuredly not have any effect on FXI inhibitors [35,36]. What remains is understanding whether the mysterious universal antidote, Ciraparantag (PER977), a small cationic molecule whose development has not seen any updates for several years, will have an effect [37].

## 3. Potential of FXI Inhibitors in Venous Thromboembolic Events

### 3.1. Physiopathology of Venous Thrombosis

Thrombosis, the formation of blood clots within venous blood vessels, primarily occurs due to the exposure of tissue factor (TF) on the surface of an injured vascular endothelium [11]. TF binds with activated factor VII (FVIIa) to form the FVIIa/TF complex, otherwise known as the extrinsic pathway [11]. This complex enables the activation of FX, which subsequently leads to the activation of thrombin [11]. This reaction is confined to the TF exposure area, and FXI plays a role in propagating intraluminal thrombus growth by aiding in the formation of the intrinsic pathway after activation by thrombin, which in turn enables the activation of FX and subsequently thrombin [11]. Cancer induces a systemic hypercoagulable state that raises the baseline thrombotic risk among affected patients. Cancer-associated thrombosis, like all venous thrombosis, is initiated by endothelial activation (e.g., catheter-related thrombosis), increased blood hypercoagulability, and blood stasis (e.g., compressive-metastasis-related thrombosis) [38]. The presence of a malignancy often amplifies the activation of the coagulation cascade, as well as the activation of blood cells, such as platelets and leukocytes [39,40,41]. Neutrophils can enhance thrombosis by releasing neutrophil extracellular traps (NETs) and by secreting proteases that target the Tissue Factor Pathway Inhibitor (TFPI) [42,43]. The reduction in the concentration of TFPI leads to increased TF activity and coagulation activation [42,43]. The contact pathway is activated by the elevation of circulating cell-free DNA and extracellular vesicles [44]. Furthermore, fibrinolysis is impaired due to an elevated concentration of Plasminogen activator inhibitor 1 [45]. In addition, cancer treatments, including chemotherapy and new targeted therapies, may increase thrombosis through mechanisms that are not yet fully understood [46]. Lastly, patient characteristics, such as age, sex, and previous history of thrombosis, as well as co-morbidities, such as anemia, renal and liver impairment, and sepsis, are also determinants of hypercoagulability in cancer patients [39].

### 3.2. Management of Cancer-Associated Thrombosis

Amongst patients with cancer-associated thrombosis, there is no differentiation with respect to the indication of anticoagulation, regardless of whether the patient has developed a pulmonary embolism or a deep-vein thrombosis. Cancer patients consistently receive treatment for a minimum duration of 6 months. (Guidelines are summarized in Table 2.) If the cancer remains active beyond this timeframe, as indicated by ongoing treatment or the presence of detectable cancer masses, the anticoagulation regimen is extended. Managing thrombosis in cancer patients presents unique challenges compared to non-cancer patients, primarily due to the delicate balance between the heightened risk of recurrent thrombosis and the elevated susceptibility to major bleeding resulting from anticoagulant therapy. In 2003, the guidelines established by the CLOT trial endorsed the use of LMWH monotherapy as the standard-of-care treatment for VTE in cancer patients [45]. With the advent of the new generation of anticoagulants, specifically direct oral anticoagulants, numerous randomized controlled trials were initiated to evaluate the efficacy and safety of this novel therapeutic class in the initial management of VTE among cancer patients. In Figure 3, the collective findings from these studies consistently affirm the favorable attributes of DOACs for the treatment of thrombosis when compared to LMWHs [46].

### 3.3. FXI Inhibitors to Prevent Venous Thromboembolic Events

A compelling way to ascertain the effectiveness of a new anticoagulant treatment in VTE is through post-orthopedic surgery prevention, given the high incidence of VTE events without thromboprophylaxis [47]. As such, several phase II randomized trials have assessed the safety and efficacy of FXI inhibitors in preventing thromboembolic events post orthopedic surgery, especially in knee surgery (Table 3) [48,49,50,51].

In a meta-analysis by Nopp et al., irrespective of the FXI inhibitor treatment (antibody or small molecule) compared to 40 mg of enoxaparin, there was a reduction in the risk of VTE with a relative risk (RR) of 0.59 (95% CI, 0.37–0.94) [2]. Furthermore, a reduced RR of developing clinically relevant bleeding in patients treated with an FXI inhibitor compared to enoxaparin was observed with 0.41 (95% CI, 0.19–0.92) [52]. A second meta-analysis published by Presume et al. corroborates these findings, with an evaluation of clinically significant bleeding, id est, an association of major bleeding and clinically significant non-major bleeding. The risk of recurrence had an RR of 0.50 (95% CI, 0.36–0.65), and, for the risk of clinically significant bleeding, an OR (odd ratio) of 0.41 (95% CI, 0.22–0.75) [53]. Of note, these trials did not include patients with active cancer, and consequentially patients with cancer-associated thrombosis. Reducing the risk of thrombus with thromboprophylaxis appears to be an option in cancer patients. Indeed, several trials in cancer patients treated with chemotherapy demonstrated that thromboprophylaxis decreases the global risk of VTE. Khorana et al. published a study in 2017 showing a non-significant reduction in the risk of thrombosis (12% on dalteparin vs. 21% on placebo, hazard ratio of 6.9) with a statistically significantly increased risk of clinically relevant bleeding, in cancer patients on chemotherapy with a Khorana score ≥ 3 after 12 weeks of prophylactic Dalteparin [54]. The AVERT and CASSINI studies evaluated FXa inhibitors in cancer patients. Despite a hazard ratio in favor of thromboprophylaxis with FXa inhibitors, they drew the same conclusions regarding the risk of bleeding [55,56]. Thus, thromboprophylaxis should be prescribed only to cancer patients at high risk of VTE [39]. For thromboprophylaxis, the promising lower risk of bleeding could lead to an interesting use of FXI inhibitors for thromboprophylaxis; however, to our knowledge, no study has been designed to explore that therapeutic option in cancer patients.

### 3.4. FXI Inhibitors to Treat Venous Thromboembolic Events

The management of VTE in cancer patients poses a unique challenge compared to non-cancer individuals, due to the delicate balance between the heightened risk of recurrences and the increased risk of clinically relevant bleeding, under anticoagulant treatment. In 2003, based on the CLOT trial, guidelines promoted low-molecular-weight heparin (LMWH) monotherapy as the primary treatment for cancer patients with VTE [48]. However, the advent of FXa inhibitors prompted several randomized controlled trials to evaluate their safety and efficacy in cancer patients. All studies concluded that FXa inhibitors are as beneficial for thrombosis treatment as LMWHs [57,58,59,60,61,62]. A meta-analysis by Planquette et al. incorporated five randomized controlled trials, i.e., the Hokusai VTE cancer, ADAM-VTE, SELECT-D, CASTA DIVA, and CARAVAGGIO trials, comparing FXa inhibitors and LMWHs [62]. The conclusion was that FXa inhibitors were as effective in preventing VTE, although an elevated risk of clinically significant bleeding with FXa inhibitors was identified. Further research is needed to evaluate the applicability of these randomized controlled trials to everyday practice, where patients with more comorbidities are often encountered [63]. Today, an increasing number of cancer patients are favoring oral treatment with FXa inhibitors as the first choice for anticoagulation, because of an expected better adherence and more convenient usage (tablets versus injections), provided their risk of bleeding permits it.

However, there are certain restrictions to the use of FXa inhibitors. International guidelines for patients with thrombosis and cancer mention a significant risk associated with the use of FXa inhibitors in patients without a high risk of bleeding. Hence, for patients with gastrointestinal cancer and urinary tract cancer, the usage of FXa inhibitors (and thrombin inhibitors) should be avoided [57,64]. Buoyed by the satisfactory results of thromboprophylaxis in non-cancer patients, phase III trials have directly commenced in the therapeutic treatment of thrombosis associated with cancer. In this population, the hemorrhagic risk and the thrombotic risk are competing and failures to demonstrate their effectiveness have been observed with all available molecules, both on the prothrombotic and hemorrhagic sides. Notably, on the hemorrhagic side the type of cancer plays a role, with a predominance of unremoved gastrointestinal and genitourinary sites in patients most at risk of bleeding. Given these findings, two trials have been designed. The first, ASTER NCT05171049, compares Abelacizumab at a dose of 150 mg with Apixaban at a dose of 5 mg twice a day in patients suffering from cancer whose tumor mass is not in contact with the mucosa. The second, MAGNOLIA NCT05171075, compares Abelacizumab at a dose of 150 mg with Dalteparin at a dose of 200 IU/kg per day for one month, followed by 150 IU/kg per day.

### 3.5. FXI Inhibitors to Prevent Catheter-Related Thrombosis

In catheter-related thrombosis, inhibition of FXI could facilitate the dissolution of the fibrin sheath conducive to thrombotic events. A common situation outside of intensive care, where patient management involves the insertion of a catheter, is dialysis. Due to kidney failure and the use of a heparinized dialysis circuit, dialysis patients are at a high risk of bleeding. Several studies using FXI inhibitors in dialysis have been conducted and published with small molecules, ASOs, and antibodies and concluded with encouraging results (Table 4) [25,65].

In a canine model of veno-venous extracorporeal membrane oxygenation (ECMO) using EP-7041, Pollack et al. demonstrated a prolongation of the aPTT and a reduction in bleeding associated with ECMO in this model [66]. Unsurprisingly, given the established interrelation between FXI activation and the presence of material, a phase II clinical trial has begun assessing Xisomab 3G3 (AB023) versus placebo for the prevention of catheter-related thrombosis in cancer patients. It is noteworthy that Xisomab 3G3 (AB023) has not been evaluated in other clinical situations requiring thromboprophylaxis.

## 4. Other Clinical Needs of Cancer Patients

### 4.1. FXI Inhibitors to Prevent Arterial Thromboembolic Events

An increasing number of studies have examined the link between VTE events and atherothrombosis, including in patients with cancer [67,68]. As with VTE, arterial thrombosis also involves exposure to TF during the rupture of an atherosclerotic plaque or its expression on activated monocytes or TF-bearing microvesicles, triggering the generation of small amounts of thrombin [68,69]. Thrombus expansion depends on the amplification of coagulation via thrombin-mediated activation facilitated by FXI, as well as thrombin itself, a potent platelet agonist [70]. The management of atheromatous diseases such as myocardial ischemia, carotid stenosis or occlusions, and lower-limb obliterating arteriopathy relies on platelet inhibition by single or dual antiplatelet treatments [71,72]. Despite this approach, up to 11% of patients suffering from an atheromatous pathology experience recurrent ischemic events each year. In these patients, it appears that suppression of platelet activity is not sufficient to avoid thrombosis [72,73]. Since thrombin has a dual effect on coagulation and platelets, this could explain the failure of antiplatelet agents. And thrombin-activated platelets are likely more frequent in cancer patient groups [74]. In the COMPASS (Cardiovascular Outcomes for People Using Anticoagulation Strategies) trial, 27,395 patients with coronary artery disease or stable peripheral arterial disease were assigned to one of three treatment arms: rivaroxaban 2.5 mg twice daily with 100 mg aspirin once daily; rivaroxaban 5 mg twice daily alone; or aspirin 100 mg once daily [75]. The primary outcome, a composite of cardiovascular death, stroke, or non-fatal myocardial infarction, was significantly lower with rivaroxaban plus aspirin than with aspirin alone (4.1% and 5.4%, respectively; hazard ratio = 0.76; 95% CI (0.66–0.86)). However, the rate of major bleeding was significantly higher in the rivaroxaban plus aspirin group than in the aspirin-alone group (3.1% and 1.9%, respectively; hazard ratio = 1.70; 95% CI (1.40–2.05)) [60]. While the rate of net clinical benefit (the occurrence of a composite of cardiovascular death, stroke, myocardial infarction, and fatal or symptomatic bleeding in a critical organ) was 4.7% in the rivaroxaban plus aspirin group and 5.9% in the aspirin-alone group (HR = 0.80; 95% CI (0.70–0.91)) is in favor of the combination, the incidence of major bleeding favors the exploration of safer solutions such as FXI inhibitors [75]. On the other hand, when we look at the included population of patients who developed cancer during follow-up, we observe that this combination is less favorable [76]. Major gastrointestinal bleeding was associated with a 20-fold increase in new diagnoses of gastrointestinal cancer (9.3% versus 0.7%; HR = 22.6; 95% CI (14.9–34.3)) and hematuria was associated with a 98-fold increase in new diagnoses of urinary cancer (14.2% versus 0.2%; hazard ratio = 98.5; 95% CI (68.0–142.7). In 66 cases, gastrointestinal bleeding also occurred for cancers of other types, with a hazard ratio of almost 2 (3.8% versus 3.1%; hazard ratio = 1.70; 95% CI (1.20–2.40)) [76]. These results suggest the importance of developing an anticoagulant able to prevent atherothrombosis in combination with an antiplatelet drug with a lower risk of bleeding for cancer patients, due to their bleeding sensitivity. In Table 5 the studies regarding arterial thrombosis in the general population are summarized. So far, no study dedicated to cancer patients exists regarding their arterial risk of thrombosis.

### 4.2. Chronic Complications of Cancer-Associated Thrombosis (Post-Thrombotic Syndrome, Chronic Thromboembolic Pulmonary Hypertension, and Recurrence)

Although we are able to effectively prevent deaths post VTE, there is currently no therapeutic arsenal to prevent sequels such as chronic thromboembolic pulmonary hypertension (CTEPH) or post-thrombotic syndrome (PTS) [77]. From a recent murine model, it appears that a reduction in macrophage activity through the inhibition of FXI would allow for a faster resolution of clot formation [78]. Catella-Chatron et al. observed that the medical evaluation of CTEPH may be lower for patients with cancer, despite some of them having a reasonable survival expectancy [79]. Direct inhibitors of FXa are about to become the drugs most used in cancer patients, but are not recommended for CTEPH patients. If FXa inhibitors are increasingly used in cancer patients, their potential impact on the development of CTEPH and their use in CTEPH cancer patients will deserve further research. A follow up of the cancer patients that developed thrombosis included in the studies with Abelacizumab and Xisomab may be very informative for understanding both CTEPH and PTS development under FXI inhibitors.

## 5. Conclusions

The pathophysiologic approach to the inhibition of FXI make it a promising target to solve the prothrombotic situations encountered with patients with a cancer-mediated procoagulant state (e.g., during chemotherapy, malignancy, or catheter placement). Subject to the results of ongoing trials, the factor XI inhibition pathway appears to be an encouraging way of improving the management of patients suffering from cancer-associated thrombosis.

## Figures and Tables

**Figure 2 ijms-24-14433-f002:**
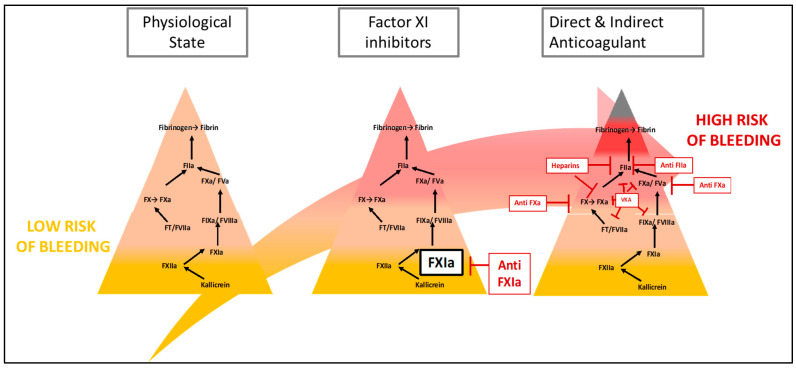
FXI inhibitors and other anticoagulants. The coagulation cascade is represented as a pyramid where the closer you get to fibrin the higher is the risk of coagulation. Therefore regarding anticoagulant the inhibition of thrombin is at a higher risk than the inhibition of FXIa. VKA stands for vitamin K antagonist. Black arrows represent the traditional pathway of coagulation.

**Figure 3 ijms-24-14433-f003:**
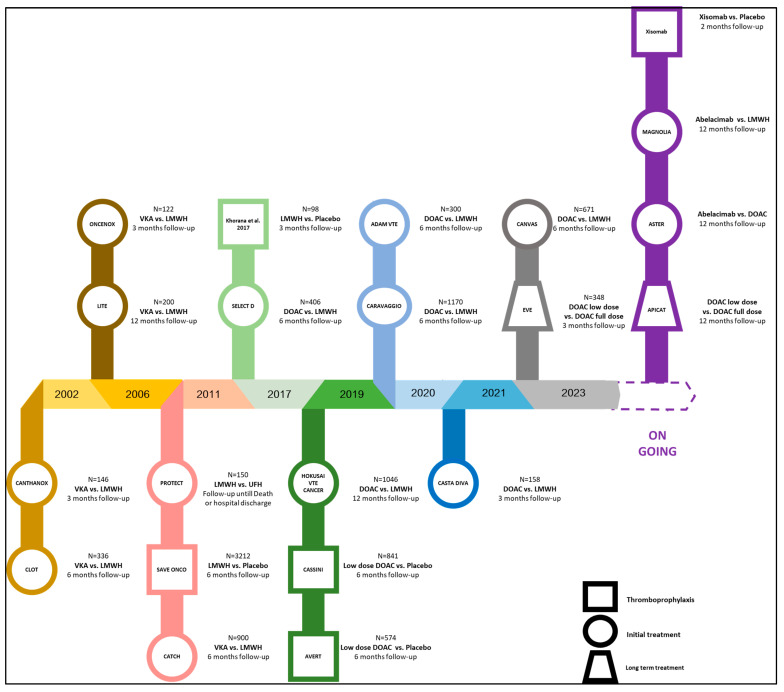
Randomized controlled trials for thromboprophylaxis treatment and secondary prevention in cancer-associated thrombosis.

**Table 1 ijms-24-14433-t001:** FXI inhibitors.

Drug Type	Molecule	Target	Mechanism	Oral Route Available	Half Life	Action	Renal Excretion	Cytochrome p450 Metabolism	Potential for Drug-Drug Interaction	Need for Reversion Strategy	Is It a Suitable Option for Cancer Associated Thrombosis
Antisense oligonucleotides or ASO	Fesomersen IONIS FXI-LRx (ISIS 416858) FXI-ASO (ISIS416858)	FXI mRNA	Specific protein synthesis blocking	No	Long: weekly administration	Slow and long acting	No	No	No	Yes: FXI replacement	The slow setting makes it less likely; however, it can be interesting for the prevention of CAT
Aptamers	/	FXIa	Specific protein binding	No	Short: daily administration	Fast and short acting	No	No	No	No	There is not enough evidence to make an opinion
Monoclonal antibodies	Abelacimab (MAA868), Osocimab (BAY1213790), FXI-175, FXI-203 (preclinical), 14E11 (preclinical), Xisomab 3G3 (AB023), MK 2060	FXI or FXI synthesis	Specific protein binding and decreases its concentration	No	Long: monthly administration	Fast and long acting	No	No	No	Yes: no existing failure of FXI replacement	Yes
Natural inhibitors	IrCPI	FXIa or FXIa + FXIIa	Specific protein binding	No	Short: daily administration	Fast and short acting	Unknown	No	Unknown	No	There is not enough evidence to make an opinion
Small peptidomimetic molecules	Asundexian (oral). Milvexian (oral)/Frunexian EP-7041a, BMS-962212, ONO-7684 (oral), BMS-654457 (preclinical), ONO-5450598 (preclinical),BMS-262084 (preclinical), SHR2285 (oral)	FXIa or FXI + plasma kallicrein	Specific protein binding	Yes	Short: daily administration	Fast and short acting	Biliary and 15% renal excretion	CYP_3A4_	Midazolam Rifampicin Verapamil, Ketoconazole…	No	The risk of interaction makes it less likely

**Table 2 ijms-24-14433-t002:** Current guidelines regarding the management of cancer-associated thrombosis.

Recommendation	Thromboprophylaxis	Intital Treatment	Long-Term Treatement
ACCP 2016	/	LMWH	LMWH or DOAC or VKA
ASCO 2019	LMWH or DOAC	LMWH, Fondaparinux or DOAC	LMWH or DOAC
ISTH 2019	Khorana ≥ 2 LMWH or DOAC Khorana ≥ 2 LMWH		
ITAC 2019	LMWH or Fondaparinux	UFH, LMWH, DOAC or VKA	
ASH 2021	LMWH	LMWH or DOAC	
French recommendations 2021	No thromboprophylaxis	LMWH	LMWH or DOAC or VKA
ACCP = American College of Chest Physicians ASCO = American Society of Clinical Oncology	ASH = American Society of Hematology ISTH = International Society on Thrombosis and Haemostasis	ITAC = International Initiative on Thrombosis and Cancer	

**Table 3 ijms-24-14433-t003:** Studies with FXI inhibitors in venous thrombosis on https://clinicaltrials.gov and https://eudract.ema.europa.eu (accessed on 20 August 2023).

Venous Thromboembolism	Study Name	Registration Number	Drug Name	Comparator	Status
Orthopedic-surgery-related thrombosis prevention	FXI-ASO TKA trial	NCT01713361	IONIS FXI-LRx (ISIS 416858) 200 mg or 300 mg SC	Enoxaparin 40 mg	COMPLETED
	AXIOMATIC-TKR trial	NCT03891524	Milvexian (BMS-986177) (JNJ70033093) oral 5 mg, 50 mg, 100 mg, or 200 mg twice daily or 25 mg, 50 mg, or 200 mg once daily	Enoxaparin 40 mg	COMPLETED
	FOXTROT trial	NCT03276143	Osocimab (BAY1213790) Single IV postoperative doses of 0.3 mg/kg, 0.6 mg/kg, 1.2 mg/kg, or 1.8 mg/kg Single IV preoperative doses of 0.3 mg/kg or 1.8 mg/kg	Enoxaparin 40 mg	COMPLETED
	ANT-005 TKA trial	EudraCT number, 2019-003756-37	Abelacimab (MAA868) 30 mg, 75 mg, or 150 mg	Enoxaparin 40 mg	COMPLETED
COVID-19-related thrombosis prevention	COVID-19 ThromboprophylaXIs	NCT05040776	Frunexian EP-7041a	Clinician choice of thromboprophylaxis	ONGOING
Cancer-associated thrombosis treatment	MAGNOLIA trial	NCT05171075	Abelacimab (MAA868) 150 mg	Dalteparin 20 mg	ONGOING
	ASTER trial	NCT05171049	Abelacimab (MAA868) 150 mg	Apixaban 5 mg bid	ONGOING

**Table 4 ijms-24-14433-t004:** Studies with FXI inhibitors in catheter-related thrombosis.

Catheter-Related Thrombosis	Name of the Study	Registration Number	Drug Name	Comparator	Status
End-stage renal disease	/	NCT04534114	Factor XI LICA (BAY 2976217)	Placebo	UPCOMING
	ESMERALD trial	NCT03358030	IONIS FXI-LRx (ISIS 416858) 200 mg, 250 mg or 300 mg	Placebo	ONGOING
	/	NCT02553889	IONIS FXI-LRx (ISIS 416858)	Placebo	COMPLETED
	/	NCT03196206	Milvexian (BMS-986177)	Enoxaparin	COMPLETED
		NCT02902679	Milvexian (BMS-986177)	None	COMPLETED
	MK-2060-004	NCT03873038	MK-2060	Placebo	COMPLETED
	MK-2060-007	NCT05027074	MK-2060	Placebo	ON GOING
	/	NCT03787368	Osocimab (BAY1213790)	Placebo	COMPLETED
	/	NCT04523220	Osocimab (BAY1213790)	Placebo	UPCOMING
	/	NCT04510987	BAY2433334 (Asundexian)	None	COMPLETED
Catheter-related thrombosis prevention		NCT04465760	Xisomab 3G3 (AB023)	Placebo	ONGOING

**Table 5 ijms-24-14433-t005:** Studies with FXI inhibitors in arterial thrombosis.

Arterial	Registration Number	Study	Drug Name	Comparator	Status
AOMI	AXIOMATIC-SSPtrial	NCT03766581	Oral Milvexian (BMS-986177) (JNJ70033093) in association with aspirin and clopidogrel	Placebo	COMPLETED
FA	PACIFIC-AF	NCT04218266	Asundexian 20 or 50 mg	Apixaban 5 or 2.5 mg	COMPLETED
Stroke	OCEANIC-STROKE	NCT04304508	Asundexian 10, 20 or 50 mg	Placebo	COMPLETED
IDM	PACIFIC-AMI	NCT04304534	Asundexian in association with aspirineand clopidogrel	Placebo	COMPLETED
	OCEANIC-AF	NCT05643573	Asundexian	Apixaban 5 or 2.5 mg	ONGOING

## Data Availability

Not applicable.

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
