# Peer review of "Inhibition of Factor XI: A New Era in the Treatment of Venous Thromboembolism in Cancer Patients?"

_ijms, 2023, doi:10.3390/ijms241914433_

Round 1

Reviewer 1 Report

The review of Poenou et al aimed to summarize the current data concerning FXI inhibitors in cancer patients. Although the review is clear and well written the focus on cancer patient is limited to only two parts of the review.

The use of ASO often leads to thrombocytopenia, this may limit the use of such technology in the field of thrombosis management: doi: 10.1016/j.thromres.2021.01.006 This can be discussed.

The part 3.1 should be extended with a specific section dedicated to thrombotic events occurring in cancer patients. In addition, the review would benefit a section summarizing the current guidelines concerning thrombotic events (first and recurrence) in cancer patients and highlighting the difference with the thrombotic events in the general population.

The part 4 is out of topic except if the authors clearly explain to the reader the importance of catheter-dependent thrombosis in cancer patients and compare this to other pathology in which such thrombosis are of major importance.

Tables 2-4 lack informative content such as the main result of each study. In figure 2, Büller et al study is numbered “42” while in the reference part it is “41”, please check references number. In table 3, is there a reason for FOXTROT trial to be highlighted in italic?

The review would greatly benefit from the presence of at least 2 scheme figures:  one about FXI in the coagulation cascade and the current inhibitors and one about FXI inhibition to manage thrombosis and current knowledge in cancer patients and comparison to other anticoagulant treatments. The presence of figures would also limit text repetitions (eg. coagulation cascade description).

Minor:

What is the contribution of each author (10 different authors is a really high number for a review)?

Spaces between number and units are often missing.

Author Response

On the behalf of all the coauthors we would like to thank you for your remarks that improve the exhaustivity of this review.

Here is attached a response point by point 

Best regards, 

Reviewer 2 Report

Dear Authors!

Thank you so much for the interesting review. Developing new anticoagulants is important especially for many patients groups that are not managed that successfully using standard protocols. The information that you presented will be of interest for those who are involved on VTE management and research.

I have some minor remarks.

Line 19. DOACs as effective as AVK and demonstrate decreased but not increased risk of bleeding. I understand what you mean. That the have risk of bleeding. But your statement may lead to misunderstanding.

Paragraph 3.1. It would be useful to add a graphic representation showing the place and the role of FXI in coagulation cascade.

Line 179. Please, add that those trials included not just cancer patients but patients on chemotherapy.

Line 182. You mean cancer patients on chemotherapy?

Lines 209-210. Both guidelines don’t include suggestions or recommendations to avoid DOACs in gastrointestinal or urinary tract cancer. They just mention significant risk. Please, correct the statement.

Section 4 is about catheter-related thrombosis which is also VTE. I recommend to add these paragraphs to section 3.

Paragraph 4.1. needs relevant references. The only ref. 16 seems to be relevant to just a last statement.

None

Author Response

On the behalf of all the coauthors we would like to thank you for you supportive review. Here is attached a point by point response

kindly

Round 2

Reviewer 1 Report

I would like to thank the authors for taking care of my comments. I have no further comments. There are only few typos in the newly added parts and figures (figure 1, FXI upside-down, figure 3 LMHW instead of LMWH) to correct in the final version.